# Exercise-Induced Bronchoconstriction in Children: State of the Art from Diagnosis to Treatment

**DOI:** 10.3390/jcm13154558

**Published:** 2024-08-05

**Authors:** Roberto Grandinetti, Nicole Mussi, Arianna Rossi, Giulia Zambelli, Marco Masetti, Antonella Giudice, Simone Pilloni, Michela Deolmi, Carlo Caffarelli, Susanna Esposito, Valentina Fainardi

**Affiliations:** Pediatric Clinic, Department of Medicine and Surgery, University of Parma, 43125 Parma, Italy; robertograndinetti93@gmail.com (R.G.); nicole.mussi1@gmail.com (N.M.); arianna.rossi@unipr.it (A.R.); zambelligi@gmail.com (G.Z.); marco.masetti@unipr.it (M.M.); antonella.giudice@unipr.it (A.G.); simone.pilloni@unipr.it (S.P.); mdeolmi@ao.pr.it (M.D.); carlo.caffarelli@unipr.it (C.C.); susannamariaroberta.esposito@unipr.it (S.E.)

**Keywords:** EIB, asthma, exercise induced asthma, exercise induced bronchoconstriction, children, EILO, vocal cord dysfunction, athletes, deconditioning

## Abstract

Exercise-induced bronchoconstriction (EIB) is a common clinical entity in people with asthma. EIB is characterized by postexercise airway obstruction that results in symptoms such as coughing, dyspnea, wheezing, chest tightness, and increased fatigue. The underlying mechanism of EIB is not completely understood. “Osmotic theory” and “thermal or vascular theory” have been proposed. Initial assessment must include a specific work-up to exclude alternative diagnoses like exercise-induced laryngeal obstruction (EILO), cardiac disease, or physical deconditioning. Detailed medical history and clinical examination must be followed by basal spirometry and exercise challenge test. The standardized treadmill running (TR) test, a controlled and standardized method to assess bronchial response to exercise, is the most adopted exercise challenge test for children aged at least 8 years. In the TR test, the goal is to reach the target heart rate in a short period and maintain it for at least 6 min. The test is then followed by spirometry at specific time points (5, 10, 15, and 30 min after exercise). In addition, bronchoprovocation tests like dry air hyperpnea (exercise and eucapnic voluntary hyperpnea) or osmotic aerosols (inhaled mannitol) can be considered when the diagnosis is uncertain. Treatment options include both pharmacological and behavioral approaches. Considering medications, the use of short-acting beta-agonists (SABA) just before exercise is the commonest option strategy, but daily inhaled corticosteroids (ICS) can also be considered, especially when EIB is not controlled with SABA only or when the patients practice physical activity very often. Among the behavioral approaches, warm-up before exercise, breathing through the nose or face mask, and avoiding polluted environments are all recommended strategies to reduce EIB risk. This review summarizes the latest evidence published over the last 10 years on the pathogenesis, diagnosis using spirometry and indirect bronchoprovocation tests, and treatment strategies, including SABA and ICS, of EIB. A specific focus has been placed on EIB management in young athletes, since this condition can not only prevent them from practicing regular physical activity but also competitive sports.

## 1. Introduction

Exercise-induced respiratory symptoms were first described in 1962 by Jones et al. [1] and then in 1968 by Fitch, who observed severe bronchial obstruction in an 18-year-old Olympic Gold Medalist swimmer after strenuous muscular activity [2]. Initially, the term “Exercise-Induced Asthma” (EIA) [3] was coined to indicate the narrowing of the airways occurring in asthmatic patients during or after physical exercise [4,5,6]. However, in 1970, the term “exercise-induced bronchoconstriction” (EIB) replaced the term EIA since the narrowing of the airways after or during physical activity could also occur in nonasthmatic patients [4,7,8,9].

The recent American Thoracic Society (ATS) Clinical Practice Guideline proposed to differentiate EIB between exercise-induced bronchoconstriction occurring in asthmatic patients (EIBa—EIB with asthma) and exercise-induced bronchoconstriction in patients without typical signs or symptoms of asthma (EIBwa—EIB without asthma) [9,10]. EIB is usually associated with bronchial hyperresponsiveness [4,10,11,12,13], a condition that refers to the tendency of the airways to constrict more easily and severely than normal airways in response to various stimuli [14]. These stimuli include individual susceptibility, type, duration, and intensity of physical exercise [15], and environmental conditions such as cold air, high atmospheric pressure, relative humidity, and pollutants [16,17]. Lung function tests performed after physical exercise can assess bronchoconstriction and demonstrate EIB [18,19].

The primary objective of this review was to explore and summarize the latest literature on the pathogenesis, diagnosis, and treatment of EIB in children and to provide practical guidance on the management of this condition with a focus on differential diagnosis.

## 2. Methods

We reviewed the literature by searching the Medline database via the PubMed interface using the following query string: (“pediat*” OR “paediatr*” OR “child” OR “youth”) AND (“exercise induced bronchoconstriction” OR “exercise-induced bronchoconstriction” OR “EIB” OR “exercise induced asthma” OR “exercise-induced asthma” OR “bronchoconstriction” OR “exertional asthma”). We restricted our search to the last ten years, from January 2014 to January 2024, with a preliminary result of 750 articles; these were screened by at least two members of the group. The preliminary analysis considered titles and abstracts and excluded those unrelated to the subject of our review. After this preliminary screening phase, we included prospective or retrospective cohort studies, randomized controlled trials (RCT), narrative and systematic reviews, and national and international guidelines. Exclusion criteria were articles in other languages than English; letters; and editorials. The analysis of the included studies has finally substantiated the various sections of this paper. A critical analysis of the studies is available in the Appendix A.

## 3. Clinical Presentation of EIB

Common symptoms include cough, dyspnea, wheezing, chest tightness, increased mucus production, heightened respiratory effort, diminished performance, increased fatigue, or a sense of reduced fitness in physically fit patients [20,21]. In children, stomachache or sore throat may also be indicators of EIB [20,22,23]. EIB usually manifests after exercise and typically starts within 15 min after 5 to 8 min of high-intensity aerobic training (>85% of maximal voluntary ventilation); EIB can also occur during exercise, with a peak after 10 min. Symptoms spontaneously resolve within approximately 60 min [21,24,25]. In asthmatic children, the time to reach maximal bronchoconstriction after exercise is shorter compared with nonasthmatic children, suggesting that the onset of EIB occurs during physical activity [20,23].

## 4. Epidemiology and Risk Factors

The prevalence of EIB ranges between 5 and 20% in the general population [25,26]; children and adolescents seem to be more susceptible to EIB with a prevalence of up to 45% [27]. This variability is influenced by demographics and geographics [26]. For example, the phenomenon has been quantified for developed Western countries with a prevalence that varies between 4.7% [28] and 12% [29]. Ethnicity has an unclear effect on the prevalence of EIB. A meta-analysis from 2018 based on 66 studies estimated a global prevalence of EIB in the general population of children and adolescents of 9%, with a higher risk (12%) in America and Asia-Pacific [30], while a previous analysis conducted in 1996 reported a 3.6-fold increased likelihood of developing EIB in Asian subjects (12%) compared with Caucasian subjects (4.5–3%) [27]. This wide variability of prevalence in children worldwide is shown in Table 1.

Despite discordant results [46,53], some data reported that females seem more prone to suffer from EIB compared with males [35,52], and this difference becomes more pronounced during adulthood, particularly after the age of 50 [54]. Environmental factors like vehicle fumes, crowdedness, and household animals [55] may also have a role in the different prevalence reported in the studies with higher rates of EIB in children living in urban environments compared with those living in rural areas [49]. A study, conducted among young competitive athletes, aged from 12 to 18, highlighted an increase in bronchial hyperresponsiveness in athletes exposed to air pollution, especially particulate matter 10 (PM_10_). It also found that air pollution exposure is responsible for greater epithelial damage and airway inflammation [56]. Another pollutant that contributes to the onset of EIB is nitrogen dioxide (NO_2_), as shown by Sanchez KM et al. Their cross-sectional pilot study demonstrated that children’s daily exposure to NO_2_ was linked to a decrease in lung function following exercise [57].

The prevalence of EIB among athletes varies depending on the studied population, with a higher occurrence observed in those engaged in endurance activities such as long-distance running, cycling, triathlon, and pentathlon [21]. According to a recent systematic review, the prevalence rate of EIB among general athletes is 23% [58], while for Olympic athletes the estimate is 8% [59]. However, in sports with elevated risk factors, such as swimming and activities exposing athletes to cold air, the prevalence varies widely, ranging from 25% to 75% [60,61]. For summer sports, the reported prevalence of EIB is 23%, and in winter sports, this increases up to 55% [62]. 

The climatic condition represents a crucial EIB risk factor. Several studies, conducted in pediatric populations, demonstrated that EIB is more severe in cold and dry environments than in humid regions with higher temperatures, including a greater reduction in forced expiratory flow in the 1st second (FEV_1_), peak expiratory flow (PEF) and maximal expiratory flow at 50% of forced vital capacity (MEF_50_), 20 min after physical activity [63]. This supports the “pathogenetic theories” of the EIB, which are based on the inspiration of cold and dry air.

EIB in endurance athletes is primarily associated with increased minute ventilation through mouth breathing (bypassing nasal filtration) and exposure to allergens and pollutants [64,65]. Regarding winter sports, the prevalence of asthma and EIB in cross-country skiers increases with age and a higher prevalence has been identified in athletes with a longer history of competitions [66,67,68]. The increased risk of EIB in subjects practicing winter sports could be attributed to the greater inhalation of cold air during training and competitions [69,70]. In physical activities performed on ice surfaces, the increased release of pollen from fossil-fueled ice resurfacing machines and the presence of ultrafine particles from polishing machines [70] on ice rinks used, for example, by hockey players [71] and figure skaters [72], can increase the risk of EIB. Swimming, traditionally considered a safe sport for asthmatic patients, may also be associated with EIB. This risk for EIB is explained by the “pool chlorine hypothesis” [62] that associates exposure to chlorine and organic chlorine products used for pool disinfection, such as trichloramine [73,74], with the development of EIB in certain subjects. On the contrary, Leahy et al. suggested that pool swimmers may have some protection against bronchoconstriction [75]. Hence, the hypothesis regarding chlorinated pools remains debated, and long-term studies are required to definitively determine whether chlorinated swimming pools play any role in the development of EIB. 

In children and adolescents with asthma, the prevalence of EIB is estimated to range from approximately 20% to 90% [13,21,38,55,76]. One study found that 46.7% of children with asthma experience symptoms of EIB, compared with only 7.4% of those without asthma [44]. The persistent inflammation in the airways of individuals with asthma makes them more susceptible to various triggers, explaining the higher prevalence of the condition in this population [9,15]. This increased prevalence is indicative of disease control, since it is more present in severe or poorly controlled asthma patients compared with those with well-controlled or mild asthma [13,21,30]. The presence of EIB in these patients often negatively affects their daily physical activity and athletic goals [44,77]. EIB can prevent children’s participation in physical activities since most patients with poorly controlled asthma are prone to experience exacerbations of symptoms after engaging in vigorous exercise [55]. Even a positive family history of asthma and obesity is involved in EIB onset as a personal risk factor. Obesity can negatively impact respiratory function, as demonstrated below.

Additional risk factors contributing to the prevalence of EIB include allergic rhinitis, a personal history of allergies, and atopic dermatitis [55]. Indeed, according to epidemiological evidence, as many as 40% of infants with allergic rhinitis suffer from EIB, especially those with persistent untreated signs and symptoms. Therefore, exploring these symptoms could be crucial for this condition, since a targeted nasal treatment might help improve EIB [19]. Studies have shown that atopy, elevated IgE levels, and sensitization to house dust mites are linked to EIB [37,78]. For instance, a Belgian study found that atopic swimmers with total IgE levels above 100 kIU/L had a higher risk of developing childhood asthma [79]. In addition, in the study conducted by Lin et al., with the aim to explore the prevalence and predictors of EIB in children with asthma, EIB was observed in 52.5% of the asthmatic children. Those with EIB were more likely to suffer from atopic dermatitis (*p* = 0.038) and had higher rates of allergy to *Dermatophagoides pteronyssinus* and *Dermatophagoides farinae* compared with those without EIB (*p* = 0.045 and 0.048, respectively) [60]. Likewise, other authors discovered a relationship between exposure to cat allergens and an increase in cough/bronchoconstriction during physical education in a large urban population of school children [17]. Therefore, atopy remains a significant risk factor among sensitized school children, even in the absence of asthma [26]. Furthermore, sensitization to outdoor and indoor allergens, as well as polysensitization rather than monosensitization, significantly increased the risk of EIB in children aged 5 to 10 years [26] (Figure 1).

## 5. Pathogenesis

The mechanisms involved in EIB are complex and not yet fully understood. Two main theories are proposed: “osmotic “ and “thermal or vascular theory” (Figure 2). Both theories might work together and are based on the bronchoconstriction and airway edema occurring during exercise [15]. 

According to “osmolar theory” (Figure 2A), bronchoconstriction is induced by the hyperosmolarity of airway cells, caused by the water loss needed to warm and humidify the cold and dry air inspired into the airways during exercise hyperventilation. Hyperosmolarity causes pulmonary hyperflux and degranulation of eosinophils and mast cells, which release inflammatory mediators such as histamine, interleukins (IL), tryptase, leukotrienes (LT), and prostaglandins (PG), which are responsible for smooth muscle contraction and mucus production in the bronchial tree [17,62,80]. Several studies have demonstrated the predominant role of cysteinyl leukotrienes (CysLTs) as bronchoconstriction mediators in bronchoalveolar lavage (BAL) of asthmatic patients [80,81].

On the other hand, the “thermal or vascular theory” (Figure 2B) consists in bronchial vasoconstriction and subsequent rapid rewarming with vasodilation following contact with cold air. As a result, vascular congestion and increased microvascular permeability lead to bronchial edema and bronchoconstriction [17,62,82]. 

EIB with asthma (EIBa) in asthmatic patients and EIB without asthma (EIBwa) in nonasthmatic patients have different underlying pathogenetic mechanisms. 

### 5.1. EIBa

In patients with EIBa, both the above-mentioned pathogenetic mechanisms (osmolar and thermal theories) represent triggers for bronchial hyperreactivity. 

### 5.2. EIBwa

In contrast, in patients with EIBwa epithelial dehydration caused by exercise is the main pathogenetic mechanism [55]. Recently, it was shown that an increasing level of damage-associated molecular patterns (DAMPs) released by the airway epithelium in response to dehydration can activate immunity cells, such as macrophages, facilitating the release of proinflammatory cytokines like TNF, IL-1b or IL-6 [55]. The epithelial damage was also demonstrated by the high level of Clara cell secretory protein 16 (CC16), a protein with anti-inflammatory and antioxidative effects produced by Clara cells, found in urine and serum after exercise in asthmatic subjects [83].

Neurological factors can also be implicated in the pathogenesis of EIB. Nerve endings around airways might be activated by the change in osmolarity or in response to other mediators, through the release of neurokinins, such as neurokinin A, and substance P, promoting bronchoconstriction and mucus production, particularly the gel-forming mucin MUC5AC [10]. 

Environmental conditions such as the presence of pollutants (especially PM_10_), low air temperature, and low air humidity have been shown to damage the airway epithelial barrier promoting bronchial inflammation and hyperresponsiveness. There is strong evidence that athletes exposed to these conditions have increased levels of epithelial damage and airway inflammation [37,63,72].

From a genetic point of view, the main genes involved in EIB are the PPT-1 gene, responsible for generating substance P, the aqueous water channel aquaporin (AQP5) gene, probably involved in airway reactivity, and the CC16 gene, from which CC16 is produced to protect airway cells [84].

## 6. Diagnosis

The gold standard for EIB diagnosis is the measurement of lung function changes provoked by physical exercise or surrogate challenges [9]. Relying only on symptoms without objective data to support EIB diagnosis may lead to inaccurate conclusions [10].

In 2019, Lammers et al. showed that clinical assessments including medical history, physical examination, and video images, were not effective in predicting accurately EIB severity in children, reporting a sensitivity of 84% but a specificity of 24% [85]. The same authors also demonstrated that the pediatrician could not evaluate the severity of EIB highlighting the need for an exercise challenge test [86]. The exercise challenge test is a noninvasive test conducted in a real-life setting, usually in a controlled environment such as a pulmonary function testing laboratory or a clinic. It contributes to understanding the underlying causes of exertional dyspnea, particularly by evaluating bronchial hyperreactivity to exercise [87].

The standardized treadmill running (TR) test is the most adopted exercise challenge test as it represents a controlled and standardized method to assess bronchial response to exercise [9]. Children should be at least 8 years of age, and their performance on the treadmill is usually more accurate if they are used to running during the physical activity they practice [87]. Before this age, as an alternative, a jumping castle for exercise challenge test can provide a creative and engaging method to assess response to physical activity, inducing a rapid increase in heart rate [88]. The clinician should take into consideration the possible bronchoprotective effect of prolonged warm-up or low-level exercise that may reduce the severity of EIB symptoms [89,90,91]. In order to avoid these confounding factors, it is necessary to reach a rapid rise in the work rate, ensuring that the exercise challenge elicits the desired bronchoconstrictive response. 

The environment is important to create standardized conditions. The exercise challenge test is usually performed in an air-conditioned room with an ambient temperature between 20 and 25 °C. Maintaining low relative humidity, ideally 50% or less, helps prevent excessive moisture in the air, and keeping the inspired air relatively dry and below 25 °C aims at minimizing potential triggers for bronchoconstriction [92]. In the TR, the goal is to reach the target heart rate (85% of maximal heart rate) while running, or minute ventilation, which should be achieved in a short period (2–3 min), progressively increasing the speed and the grade of the steepness of the treadmill. Once the patient reaches the target heart rate, this level of intensity should be maintained for at least 6–8 min [92,93,94,95,96]. If possible, measuring ventilation provides a more accurate assessment of the respiratory response than achieving the target heart rate, allowing for better evaluation of bronchial reactivity [92]. The target ventilation is often set at 60% of the predicted maximum voluntary ventilation (MVV), and MVV is estimated as FEV_1_ multiplied by 40 [96,97]; this calculation helps establish a level of respiratory challenge that is sufficient to elicit bronchial responsiveness. In this case, the patient has to breathe through a mouthpiece or a mask that provides valuable data on how the respiratory system responds to exercise [92]. 

After the exercise, the patient stops and FEV_1_ is measured by spirometry. The standard practice is to measure FEV_1_ at specific time points after exercise, typically at 5, 10, 15, and 30 min, but it can be tailored to patients’ needs; more frequent measurements can be required if a serious response is expected [9,98]. The assessment of EIB severity consists of comparing pre-exercise FEV_1_ basal value with the lowest FEV_1_ value recorded at the specific intervals (5, 10, 15, and 30 min); this difference is then expressed as a percentage of the pre-exercise value [9].

EIB is diagnosed when FEV_1_ decreases ≥ 10% compared with pre-exercise baseline value [8,9,10,93,96] and is classified as mild when the decrease is ≥10% but <25%, moderate when ≥25% but <50%, and severe when the fall is ≥50% [9].

For severe EIB, there could be a further division into steroid-naïve patients (decrease in FEV_1_ after exercise ≥ 50%) and steroid-treated patients decrease in FEV_1_ after exercise ≥ 30%) [98]. Details and steps to execute the exercise challenge test on the treadmill, including suggested environmental conditions, are reported in Table 2.

Absolute contraindications to the execution of the TR test are fever, heart diseases such as pericarditis and myocarditis or uncontrolled heart failure, dyspnea at rest, SpO_2_ < 85% in air or predicted FEV_1_ < 50%, uncontrolled hypertension or diabetes, acute kidney disease or acute hepatitis, recent pneumothorax/pneumomediastine, recent thoracic surgery. For a correct interpretation of the results, some medications (short-acting beta_2_-agonists (SABA), long-acting beta_2_ agonists (LABA), theophylline, ipratropium or leukotriene receptor antagonists) or some foods (coffee, tea, chocolate, stimulant drinks) must be avoided in the hours before the execution, as they can alter the parameters evaluated. Table 3 reports contraindications to test execution and medications and foods to be avoided before the challenge.

Indirect bronchoprovocation testing may represent a diagnostic option to establish EIB diagnosis [99]. Indirect bronchial provocation tests use external stimuli like dry air hyperpnea (exercise and eucapnic voluntary hyperpnea) or osmotic aerosols (inhaled mannitol) to cause the endogenous release of bronchoconstrictor mediators from airway inflammatory cells.

The eucapnic voluntary hyperventilation (EVH) test is a well-known tool for exercise-induced bronchoconstriction diagnosis but data on its feasibility in children remain limited.

Previous studies have shown that subjects generally tolerate well EVH testing but only a minority of children can actually reach the target minute ventilation during the test [92]. Burman et al. [100], instead, demonstrated its feasibility in 10-to-16-year-old children, suggesting that even reaching a minimum of 70% of target minute ventilation volume may be considered as an acceptable performance. 

Provocation test with mannitol is a safety challenge test to detect EIB [101]. Mannitol has a high specificity for asthma by causing smooth muscle contraction via the secretion of endogenous mediators, including histamine, prostaglandins, and leukotrienes [102].

Inhaled mannitol can be regarded as a safe and standardized test, but a less sensitive tool for the diagnosis of asthma in children [103]. Discomfort and cough during the test occur frequently, but only in a minority of cases lead to test discontinuation. Headache, pharyngolaryngeal pain, and cough are the most frequent adverse events [103]. However, the test holds several advantages compared with other existing tests: there is no need for additional equipment (i.e., a nebulizer) besides a spirometer, it requires no cleaning, and has only one standard operating protocol [103].

## 7. Role of Cardio-Pulmonary Exercise Testing (CPET)

CPET can be considered in the following cases: (1) assessment of dyspnea of unknown origin (i.e., with inconclusive respiratory function test); (2) differential diagnosis between pulmonary dyspnea and cardiac dyspnea; (3) functional assessment in chronic lung disease; and (4) follow-up in rehabilitation/retraining program.

CPET identifies the anaerobic threshold (AT) and measures the maximum oxygen consumption (VO_2_ max). In a deconditioned subject, a reduced anaerobic threshold (<50%) can be observed; alternatively, the diagnosis can be suspected with a reduced VO_2_ max (<80%) at the time of onset of symptoms in association with an indicator of poor physical condition like reduced AT, reduced SpO_2_, excessive increase in heart rate in response to a given workload [104].

CPET is usually performed on a treadmill or cycle ergometer, continuously detecting the inspiratory and expiratory flows, oxygen consumption (VO_2_), CO_2_ production (VCO_2_), electrocardiogram (ECG), and pulse oximetry (arterial pCO_2_ is usually evaluated noninvasively by measuring the end-tidal CO_2_ in the expired air). The “incremental protocol” is one of the most frequently applied, providing a gradual increase in the workload every 1–3 min. The test will go on until (1) muscle exhaustion (test limited by the symptoms); (2) reaching the maximum predicted cardiac frequency (maximal test); or (3) onset of signs or symptoms that recommend the suspension of the test (significant dyspnea, hyperventilation disproportionate to the workload, oxygen desaturation < 85%, ECG alterations). Individualized workload increment protocols are usually applied according to age or comorbidities [105].

## 8. Differential Diagnosis

Dyspnea, cough, wheezing, and chest pain are not exclusive to EIB since other conditions may present with similar manifestations [17]. Table 4 describes the commonest alternative diagnosis to EIB and Figure 3 is the proposed flow-chart for diagnostic work-up.

### 8.1. Exercise-Induced Laryngeal Obstruction (EILO) 

EILO is a specific type of laryngeal disorder characterized by inappropriate closure or narrowing of the vocal folds and/or supraglottic structures during exercise [40,106,107,108,109,110].

The real prevalence of EILO in the general population is still unclear. Although conflicted reports on gender differences are reported in the literature [108,111], higher prevalence is described in adolescents and young adults [112], particularly girls [40,113]. Overall, the reported prevalence among the studies is between 5 and 10% [40,108,111,113].

The closure or narrowing of the vocal folds and/or supraglottic structures during exercise leads to symptoms such as breathing difficulty, stridor, coughing, chest pain, and throat tightness during physical activity [112]. Symptoms often occur during high-intensity exercise when ventilation demands are at their peak, typically during the inspiration phase. The characteristic stridor, a wheeze or whistle sound on inspiration, is a hallmark feature and symptoms usually subside within a few minutes of stopping exercise unless there is ongoing hyperventilation [108,114].

In contrast, asthma and EIB typically present with expiratory breathing difficulties, and wheezing is more prominent after exercise rather than during it. Lung function changes indicative of asthma or EIB are typically measured in the postexercise phase, usually peaking around 3 to 15 min after exercise cessation [67,108].

Understanding these distinctions is crucial for accurate diagnosis and appropriate management of these different respiratory conditions. EILO and EIB can coexist, and this sometimes explains the persistence of breathing problems during exercise despite excellent asthma therapy [41,114].

The gold standard examination for diagnosis is continuous laryngoscopy during exercise (CLE) [112,113,115,116,117]. This procedure consists in visualizing the laryngeal structures using a flexible laryngoscope while the patient exercises on a treadmill or stationary bicycle. The ear–nose–throat (ENT) specialist can then observe real-time changes in the supraglottic and glottic structures throughout the exercise session. During CLE, the patient wears a headband or helmet to secure the flexible laryngoscope in place, with the scope inserted through the nose to provide a clear view of the larynx. Simultaneously, cardiopulmonary data are collected with the aim of reproducing symptoms of EILO during maximum level of exertion [108,115,116,118].

Spirometry can be useful but not conclusive because patients with EILO often have normal flow volume loops with no signs of obstruction [119]. 

### 8.2. Obesity

Obesity can negatively impact the respiratory system [120] but the threshold where obesity starts to harm lung function and physical performance is not clearly defined, as it may differ significantly among individuals [121,122].

In children, the relationship between asthma, physical fitness, and overweight is complex and not completely understood [122,123]. 

The correlation between obesity and asthma arises from the existing definition of obesity, which is primarily based on body mass index (BMI) value [124]. Quételet, who first proposed the formula for BMI, aimed to describe the average individual but excluded children and women from his studies [125]. BMI does not differentiate between lean and fat mass and it does not provide details on the distribution of fat deposits, which could be crucial for understanding various diseases [124]. 

In adults, a higher BMI is associated with lower lung volumes and preserved FEV_1_/forced vital capacity (FVC) ratio, suggesting that obesity results in a restrictive lung deficit [126,127]. On the other hand, several studies performed in children have found that increased BMI is linked to normal or even higher FEV_1_ and FVC values, but low FEV_1_/FVC ratio, suggesting the presence of either an obstructive lung defect or airway dysanapsis, a mismatch between the growth of lung parenchyma and the size of the airways [128,129,130]. Dysanapsis is particularly present in obese boys [128,129].

In a cross-sectional study, Ferreira et al. [122] showed that the obese group had lower FEV_1_ compared with the control group, showing that 36.8% of them had airway obstruction. Even if individuals in the obese group walked shorter distances in the 6-Minute Walking Test and exhibited lower values in some lung function indicators, no association was found between obesity and these test outcomes suggesting that obesity is not the only reason for compromised lung functionality. Musculoskeletal limitations and cardiovascular factors may also contribute to the reduced fitness level in obese subjects.

Ozgen et al. [131] found that obese children exhibited lower functional exercise and lung capacities compared with nonobese children. The study by Van Veen et al. [132] demonstrated that BMI-z-score was strongly associated with EIB in asthmatic boys, even in children with normal weight, and that the severity of EIB was significantly higher in overweight or obese children. Furthermore, Forno et al. [133] showed how obesity and overweight were harmful to lung function with lower lung volumes, especially in patients without asthma. Several potential mechanisms have been proposed to explain this association [132]: truncal adiposity can exert pressure on the diaphragm and chest wall, reducing lung volumes and functional residual capacity; metabolic dysregulation, including insulin resistance, dyslipidemia, and altered adipokine secretion can have systemic effects on inflammatory pathways and immune responses, which may influence airway inflammation; secretion of various proinflammatory cytokines and adipokines, such as leptin, can contribute to systemic inflammation and may promote airway hyperresponsiveness [134]; and airway dysanapsis may contribute to airflow limitation, increased airway resistance, and potentially exacerbate asthma symptoms [130]. 

In a more recent study, Malmberg et al. [135] evaluated the association between BMI and EIB in a large sample of children with respiratory symptoms and they found that higher BMI was associated with lower exercise performance and increased perception of respiratory symptoms, such as coughing and dyspnea, but not with EIB or exercise-induced wheeze. The absence of a direct association between BMI and EIB or exercise-induced wheeze confirms that other factors, such as airway inflammation or hyperresponsiveness may play a more significant role in these specific outcomes.

On the other hand, Souza de Almeida et al. [136] demonstrated that obese asthmatic children were at increased risk of experiencing bronchoconstriction, particularly those with moderate or severe asthma, with a more delayed recovery than their nonobese peers.

An increase in weight and overgrowth are further influencing factors. Den Dekker et al. [137] revealed in their meta-analysis that higher birth weight and rapid infant weight gain correlated with increased FEV_1_ and FVC but decreased FEV_1_/FVC ratio in school-age children. In contrast, Strunk et al. found that compared with participants who were never obese (BMI 23.4 ± 2.6 kg/m^2^), individuals who became obese after the trial (BMI 31.5 ± 3.8 kg/m^2^) experienced significant declines in FEV_1_ and FEV_1_/FVC between 26 and 30 years of age, with no notable associations with FVC [138]. 

Furthermore, obese children may exhibit obesity-related issues and conditions such as gastroesophageal reflux disease (GERD) and chest construction that might lead to dyspnea, which could be misinterpreted as asthma [123].

Further studies in children with EIB and overweight or obesity are essential for developing evidence-based strategies for preventing and managing both conditions effectively. 

### 8.3. Physical Deconditioning

The beneficial effects of moderate or moderate-intense daily physical activity (at least 1 h per day) for children and adolescents are universally recognized, as evidenced by countless scientific evidence and from the latest World Health Organization (WHO) guidelines [139]. When approaching physical deconditioning in terms of differential diagnosis with EIB, we must consider how limitation to the effort is a physiological phenomenon and that is the commonest cause of dyspnea in children and adolescents [140,141]. Effort limitation can happen also in physically trained subjects, because much depends on the general health status of the subject, in particular cardiovascular and pulmonary conditions. However, effort limitation occurs earlier in subjects with significant physical deconditioning [140,141]. Considering the alarming data on overweight and obesity prevalence [142,143], often associated with sedentary behaviors, deconditioning should always be excluded when dealing with breathlessness during physical exercise. 

The phenomenon of deconditioning results in the gradual cessation of physical activity with worsening performance, dyspnea, or other associated symptoms. In addition, metabolic changes due to poor physical training can result in a proinflammatory state due to increased adipose tissue, the release of cytokines, and decreased muscle mass, thereby aggravating some lung and cardiovascular conditions [144,145,146].

In this case, differential diagnosis is a diagnostic challenge since many diseases can manifest with reduced tolerance to effort and physical activity. In bronchial asthma with suboptimal control, but also in the initial phase of heart disease, a lower capacity of practicing physical exercise can be the first clinical sign. Personal history for syncope, SpO_2_ below the expected limits for age, fever, chest pain, and family history of heart disease should be investigated. If deconditioning is suspected, stress tests in particular environmental conditions such as cold air or other stimuli like methacholine challenge can be performed. 

CPET remains a cornerstone in making the differential diagnosis between functional physical limitation and physical deconditioning. As already mentioned, a deconditioned subject will have a reduced anaerobic threshold (<50%) or a reduced VO_2_ max (<80%) at the time of onset of symptoms. These can be associated with other indicators of poor physical condition, for example, an excessive increase in heart rate in response to a given workload; on the other hand, a subject with a physiological limitation will have a normal or elevated VO_2_ max.

CPET is a very important test for patients with EIB and asthma and helps to prevent from escalation of asthma treatment when unnecessary. A recent review of 49 CPETs performed by children with asthma demonstrated that EIB causes symptoms during exercise only in a small percentage of cases, while in almost one-third of cases, physical deconditioning is the underlying cause of exercise limitation [147]. 

A case-control study evaluated the underlying etiology of breathlessness in obese adolescents with or without asthma and found that breathlessness and symptoms during physical activity were primarily due to poor physical fitness and deconditioning without significant differences between obese/asthmatic and obese/nonasthmatic subjects. Interestingly, pulmonary capacity and spirometry at baseline and at the peak of exercise were similar in the two groups but the main difference with the control group was seen for the greater decrease in the VO_2_ max at the peak of CPET in the obese children regardless of the condition of asthma [148]. As said, a VO_2_ max <80% is one of the major indicators of deconditioning, preferentially in association with other indicators of poor physical condition.

These data were confirmed in a recent cross-sectional study, which analyzed asthmatic children in comparison with healthy controls (healthy controls n = 446, asthma subjects = 144) where reduced VO_2_ max and VAT were significantly altered in asthmatic subjects. The multivariable analysis confirmed that higher BMI was a predictor of impaired cardiopulmonary fitness [149].

Deconditioning was also considered during the COVID-19 pandemic as the cause of lower results during CPET in healthy children and adolescents stopped from physical activity [150].

In conclusion, deconditioning must always be considered in the differential diagnosis of EIB. The treatment for EIB-like symptoms caused by normal physiologic limitation and physical deconditioning in the absence of any underlying disease is reassurance and counseling. Appropriate retraining programs can be considered to improve symptoms and assess improvements in sport performance [141].

### 8.4. Rare Metabolic and Neuromuscular Diseases

Thanks to modern screening strategies, the diagnosis of metabolic and neuromuscular diseases generally occurs at a very early age. However, in rare cases, dyspnea and breathlessness can be symptoms of metabolic (glycogenosis and storage diseases, mitochondrial disorders, and enzyme deficits) and neuromuscular diseases (myasthenia gravis, demyelinating diseases, and motor neuron disease).

These diseases can have in common easy fatigue but also muscle cramps at the very beginning of physical efforts (due to lactic acidosis for mitochondrial disorders and deficient signal transmission in neuromuscular diseases) [87].

In the case of myasthenic syndromes, the increase in fatigue and symptoms over the day, with a peak in the evening hours, can call for subsequent specialist investigations.

The prevalence of EIB or asthma in these conditions is unknown. Lung function must be monitored to assess the progression of the disease; when possible, CPET can be performed [151].

In myasthenic syndromes, the use of salbutamol can be considered to improve dyspnea and muscle strength [152].

## 9. The Point of View of Cardiologists

As already mentioned, stress intolerance can also be associated with the initial phase of cardiac disease.

When first-level investigations like spirometry are inconclusive, the exercise challenge test may be helpful in differentiating pulmonary dyspnea from a possible cardiac cause. However, as evidenced by many analyses, this additional test is not always conclusive (due to poor standardization or poor reproducibility) [153]. A standardized CPET is usually crucial in the diagnostic algorithm [154].

Among the cardiac diseases, hypertrophic cardiomyopathy (HC) and effort-induced ventricular (EIVT) or supraventricular tachycardia (SVT) can be confused with EIB.

### 9.1. Hypertrophic Cardiomyopathy (HC) 

In HC dyspnea is a common symptom of the disease, particularly during physical exercise due to the augmented obstruction of the left ventricular outflow tract. Clinical history and other associated symptoms like chest pain, syncopal episodes, and vertigo can support the diagnosis, which is based on echocardiography. Primitive or familiar forms are the most prevalent, but HC can also be the expression of neuromuscular diseases, metabolic disorders, or specific syndromes. While moderate and constant physical activity is recommended, competitions at high levels of intensity must be avoided for the risk of fatal arrhythmias and sudden death.

### 9.2. Effort-Induced Ventricular (EIVT) or Supraventricular Tachycardia (SVT) 

Paroxysmal tachycardias, in most cases, occur in the absence of heart disease and may represent an unexpected event during ergometric testing for the assessment of sport fitness. Physical exertion, by increasing sympathetic activity, can have a favoring effect on tachycardia, especially supraventricular tachycardia. In addition to palpitations, dyspnea during exercise is a common symptom. Hemodynamically, most of these forms are well tolerated, except in cases of very high heart rates that can result in circulatory decompensation. A complete cardiac work-up must be performed, especially in the case of a positive family history.

## 10. Treatment

### 10.1. Pharmacological Strategies

#### 10.1.1. Short Acting Beta_2_ Agonists

The acute episode of EIB follows the international guidelines for asthma treatment [155]. For the prevention and treatment of EIB, SABA is the first-choice medication [156]. Recognized by multiple meta-analyses [157], SABA administered before exercise is considered effective and safe in preventing EIA [158]. SABA must be taken a maximum of one hour before exercise, preferably 15 min before [159].

Current guidelines suggest taking SABA approximately 10–15 min before starting physical activity [9]. Preventive medication with 200 mcg (2 puffs) is usually effective [158,160] but the dose can be increased to 400 mcg (4 puffs).

Excessive use of SABA (particularly in athletes who practice frequent training) can cause tachyphylaxis and loss of the bronchoprotective effect during physical exercise [156], which is probably caused by desensitization of the beta_2_ receptors on mast cells and airway smooth muscle [161]. 

#### 10.1.2. Inhaled Corticosteroid (ICS) and Leukotriene Receptor Antagonist (LTRA)

Therefore, for patients with EIB who continue to have symptoms despite using inhaled SABA before exercise, or who require inhaled SABA daily or more frequently, it is recommended to start daily ICS or daily LTRA [9]. Regarding ICS, their effectiveness as specific therapy for EIB is dependent on the duration of therapy and dosage [157]. The short-term treatment (for 14 days) with ICS and LTRA in children and adolescents with EIB decreases airway hyperresponsiveness to exercise [162]. Stelmach et al. have also demonstrated the effectiveness of prolonged treatment with ICS alone or in combination with LTRA or LABA, especially in reducing the onset of clinical symptoms after physical exercise induced by an exercise treadmill challenge in a group of children with asthma [161]. Visser et al. demonstrated that taking a single dose of ICS before exercise was effective in preventing EIB in a population of children with mild asthma [163]. Furthermore, in one of the most recent studies, the single administration of ICS (beclomethasone dipropionate 200 ug) appears to be predictive of the effectiveness of a more prolonged treatment (beclomethasone dipropionate 100 ug twice a day for 4 weeks) [164]. However, as regards LTRAs, they are less effective than beta_2_ agonists [165] but do not develop tolerance. LTRAs are effective in preventing EIB two hours after intake and up to 24 h later [166,167].

Alternatively, other preventive medications such as mast cell stabilizing agents (like sodium cromoglycate and nedocromil sodium) and inhaled anticholinergic agents (like ipratropium bromide) before exercise can be considered [9].

#### 10.1.3. Combination of ICS and Long-Acting Beta_2_ Agonists (LABA)

An effective medication is a combination of ICS/LABA, used as needed. In particular, preventive medication with Budesonide/Formoterol with a single inhalation 5 to 20 min before exercise, in a population with mild asthma aged ≥12 years, has proven to be more effective than premedication with SABA in preventing EIB. The maximum post-exercise FEV_1_ falls after 6 weeks of treatment was smaller in the budesonide/formoterol on-demand group but greater in the terbutaline on-demand group. Interestingly, EIB was reduced by 28.5% in the budesonide/formoterol on-demand group and increased by 8.9% in the terbutaline on-demand group [168]. In addition, inhalation of ICS/LABA (Budesonide/Formoterol 200/6 μg) as needed was not inferior to regular treatment with ICS alone (Budesonide 400 μg), despite a lower dose of the steroid [169]. 

Some studies have shown that treatment with Fluticasone Furoate/Vilanterol (92/22 μg) once a day for 30–60 days resulted in preventive effect against EIB in adolescents with asthma and EIA [168].

Some athletes who respond poorly to ICS-only therapy often improve after starting ICS/LABA therapy [170].

### 10.2. Nonpharmacological Strategies

#### 10.2.1. Behavioral Modifications 

Some behaviors can prevent the onset of EIB. 

Warm-up exercise: It is recommended to practice warm-up exercises 10–15 min before carrying out physical activity [9,17]. Warming up at high-intensity intervals or a combination of low and high intensity 10–15 min before exercise seems to be particularly effective to induce a refractory period of approximately two hours [9]. 

Face mask: Breathing through the nose during exercise can contribute to humidification and warming of the inspired air reducing the risk of EIB [25]. Similarly, the use of a face mask in cold weather can reduce the risk of EIB [9]. 

In addition, some studies have shown that regular continuous aerobic exercise benefits asthmatic patients on FEV_1_, PEF, FVC, and forced expiratory flow (FEF)_25–75%_ and, in general, improves patients’ symptoms and quality of life [17]. 

#### 10.2.2. Environmental Control

Environmental control is also particularly significant in the prevention of EIB. 

##### Avoidance of Cold Climates and High-Pollution Areas

It is sensible to avoid physical activity in cold climates or in polluted environments. When this is not possible, the routine use of a device (for example a mask) that warms and humidifies the inspired air during exercise is recommended [9]. 

##### Smoking and Diet

Finally, a healthy lifestyle helps to reduce the onset of EIB: exposure to smoking (both active and passive) must be avoided and a healthy diet (for example, preferring a diet rich in vitamin D and omega 3 fatty acids and low in salt) is suggested [9,145]. At least 60 min per day of moderate to vigorous physical activity is strongly recommended [131]. 

In conclusion, regarding the management of EIB, no recent meta-analyses comparing the various pharmacological treatment options available emerge from the literature. Treatment of EIB must be as individualized as possible, taking into account the current guidelines for the management of asthma, the severity of the disease, and the frequency of physical activity. A step-by-step approach appears to be the most suitable, as suggested by international guidelines for asthma. 

A summary of treatment for EIB, both pharmacological and nonpharmacological is shown in Table 5.

#### 10.2.3. Treatment of EIB in Young Athletes

Despite the scarcity of data in children, some of the indications used for adults can be reasonably extended to younger ages. Children with chronic diseases must practice sports like their peers when in good clinical conditions. In case of the progression of the disease, sport can be replaced with specific rehabilitation programs tailored to the disease and its level of severity. 

Up to 8% of Olympic athletes suffer from asthma [171,172] suggesting that this chronic disease is not a limitation if asthma is under control. Two recent studies also revealed a high prevalence of asthma among elite athletes [173] and Olympic athletes with intellectual disabilities [174], higher than in the general population.

The medications used for asthma treatment have been reviewed by the World AntiDoping Association (WADA), which has released guidelines regarding their eligibility or exclusion during competitions. 

According to current indications, athletes suffering from documented asthma can take ICS and inhaled beta_2_ agonists; the use of systemic steroids, systemic beta_2_ agonists, and other adrenergic medications is subject to authorization [175]. In particular, some concerns about performance enhancement were first noticed for some beta_2_ agonists, but salbutamol (maximum dose 1600 mcg/24 h and 800 mcg/12 h), formoterol (maximum dose 54 mcg/24 h), and salmeterol (maximum dose 200 mcg/24 h) are now permitted without authorization [176].

For drugs that need to be authorized, athletes must exhibit clinical signs of asthma and bronchial hyperreactivity, which must be confirmed by a bronchial challenge test [175]. However, for the athlete who has a proven and real clinical need for medications listed as prohibited to obtain asthma control (or medications allowed but beyond the maximum allowed dosage), an exemption for therapeutic use even during competitions can be considered [175].

Systemic glucocorticoids are prohibited in competition and a particular authorization is needed for their use outside of competition. Leukotriene receptor antagonists (montelukast, zafirlukast) and omalizumab are instead permitted agents [175].

It should be remembered, however, that the aforementioned indications apply only to adult athletes and that further studies are needed to extend these indications also in children.

## 11. Follow-Up

Data concerning the evolution over time of EIB are scarce, but this condition may increase in prevalence from childhood to adulthood. The study by Johansson et al. [177] found an increase in the prevalence of self-reported exercise-induced airway symptoms (including wheezing and dyspnea) following adolescents for a five-year period into early adulthood. The prevalence of these symptoms increased by two-fold, especially in females [177].

The evolution of EIB in athletes is also not completely clear. Data suggest that asthma and airway inflammation are only partially reversible and are influenced by the interruption of highly demanding training. In particular, athletes who continue their intensive training tend to have persistence or even an increase in airway inflammation and asthmatic symptoms. On the other hand, bronchial hyperresponsiveness seems to reduce in athletes who stop their high-intensity training [12].

The physical activity pattern practiced by children affected by EIB must be strictly monitored. There is no need to avoid physical activity in children with EIB but the child and the caregivers must be adequately instructed on EIB prevention and management of a possible acute episode of EIB. However, children with EIB are involved in less physical activity than their healthy counterparts, and the activity in these children is also reduced in terms of intensity [178]. 

Clear indications on long-term management and follow-up (repetition of exercise challenge test, adjustment or interruption of therapy) of children with EIB are currently lacking. Follow-up visits can be performed every 3–6 months as suggested for asthma management. Patients should be questioned about EIB symptoms (including occurrence, characteristics, and duration) and the efficacy of treatment. Adherence to treatment must always be assessed. Objective measurement of lung function with spirometry should be performed at every visit, while the exercise challenge test can be repeated depending on symptoms and, when needed, to assess the effectiveness of treatment [9]. Data on the safety of long-term therapies with SABA to treat EIB are poor, but it is well known that regular use of SABA in patients with asthma can desensitize the airway beta_2_ receptors reducing the bronchodilator response, making it less effective over time with a phenomenon known as tachyphylaxis; in addition, regular use can promote rebound of airway responsiveness after cessation of the treatment [179,180,181]. When symptoms are poorly controlled despite optimal adherence to treatment, alternative diagnosis should be considered as explained in the specific section of this document.

Discontinuation of treatment can be considered, but the exact timing is still a matter of debate. As for asthma treatment, the step-down strategy can be applied, and after a period of wellbeing, an attempt at suspension can be made. However, future studies are needed to establish strategies for follow-up and management of EIB.

## 12. Conclusions

EIB can be very common in patients with asthma but can also be present in people without this condition. A specific diagnostic work-up, including history, symptoms, and diagnostic tests, is crucial to identify EIB and exclude other diseases. Objective tests like the exercise challenge test are mandatory to start regular treatment with medications. Treatments should include both behavioral and pharmacological approaches according to asthma guidelines. SABA as needed before exercise is usually effective in the control of symptoms. However, EIB can be a sign of poorly controlled asthma. Follow-up and reassessments are needed over time, but clear indications on when and how to discontinue treatment are still lacking. EIB in athletes deserves particular attention since asthma per se is not a contraindication to physical activity but can prevent children and adolescents from regular practice. Regular physical activity is essential for children of all ages and, when present, EIB must be considered and treated. In summary, while significant progress has been made in understanding EIB, gaps remain in optimal diagnostic protocols and long-term management strategies. Future research should focus on large-scale, longitudinal studies to address these gaps.

## Figures and Tables

**Figure 1 jcm-13-04558-f001:**
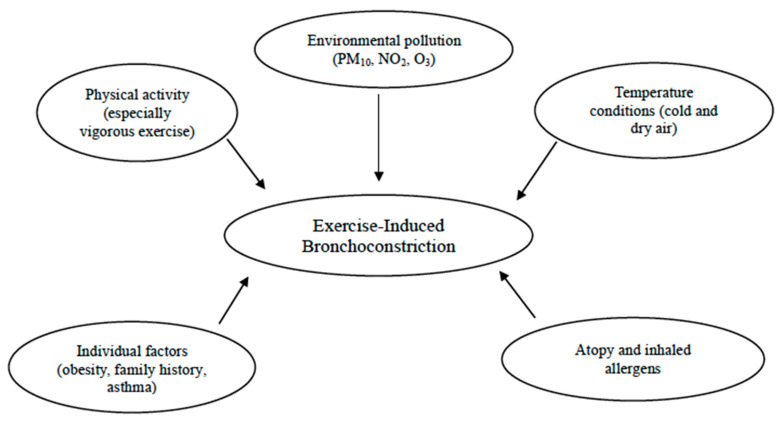
Risk factors for exercise-induced bronchoconstriction (EIB).

**Figure 2 jcm-13-04558-f002:**
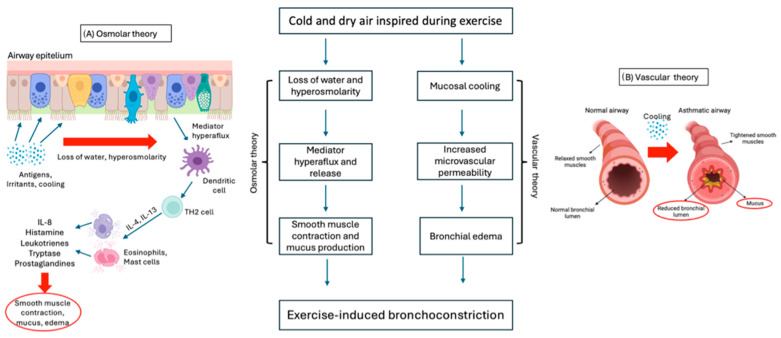
Osmotic (**A**) and vascular (**B**) theories underlying the pathogenesis of EIB.

**Figure 3 jcm-13-04558-f003:**
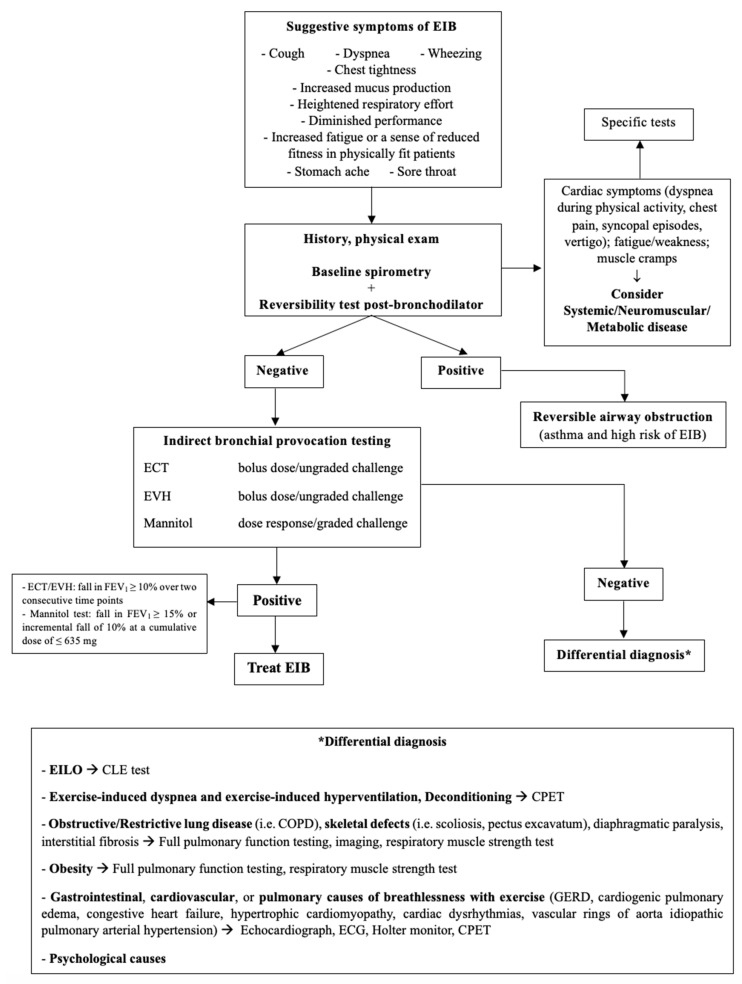
Diagnostic flowchart for EIB. EIB: exercise-induced bronchoconstriction; ECT: exercise challenge test; EVH: Eucapnic Voluntary Hyperpnea; EILO: Exercise-Induced Laryngeal Obstruction; CLE: Continuous Laryngeal Exercise Test; CPET: Cardio-Pulmonary Exercise Test; COPD: Chronic Obstructive Pulmonary Disease; GERD: Gastro-Esophageal Reflux Disease; ECG: electrocardiogram.

**Table 1 jcm-13-04558-t001:** Prevalence of EIB in children worldwide by region.

Region	Range of Prevalence	Countries	Study, Country [Reference]
Europe	From to 3% to 23%	Greece, Denmark, UK, Wales, Sweden, Poland, France, Belgium, Czech Republic	Anthracopoulos 2012—Greece [31] Backer 1992—Denmark [32] Burr 2006—UK [28] Powell 1996—UK [33] Jones 1994—Wales [34] Barry 1991—South Wales [29] De Baets 2005—Belgium [35] Caillaud 2014—France [26] Raherison 2007—France [36] B 2016—Poland [37] Cichalewski 2015—Poland [38] Vacek 1999—Czech Republic [39] Johansson 2015—Sweden [40]
Northern America	From 6% to 22%	USA, Canada	Heaman 1997—USA [41] Randolph 1997—USA [42] Vacek 1999—Canada [39] Hemmelgarn 1997—Canada [43]
Southern America	From 7.4% to 19%	Brazil	Correia Kunior 2017—Brasil [44]
Africa	From 4% to 58%	Kenya, Ghana, Algeria, South Africa	Addo-Yobo 2007—Ghana [45] Benarab-Boucherit 2011—Algeria [46] Calvert 2010—South Africa [47] Mtshali 2009—South Africa [48] Ng’ang’a 1998—Kenya [49]
Asia	From 12% to 54%	Korea, India, Japan	Sudhir 2003—India [50] Hong 2004—Korea [51] Shinohara 2019—Japan [52]
Oceania	From 10% to 15%	New Zealand	Barry 1991—New Zealand [29]

**Table 2 jcm-13-04558-t002:** Standardized treadmill running test: details on preparation and execution.

Preliminary Assessments and Environmental Settings
Check standard environmental conditions: temperature (20–25 °C), humidity (≤50%), air quality (minimal pollutants) Informed consent signed by at least one parent Monitor heart rate and SpO_2_ Nose clip
**Procedure**
Avoid physical exercise in the previous 6 h Avoid eating for 4 h prior the test Perform basal spirometry (a basal FEV_1_ of at least 75% is required) Adjust speed and grade of the treadmill to achieve within 2–3 min: 1) 85% (95% in children and élite athletes) of the maximum heart rate (220—age in years or 208—0.7 × age in years) Maintain this level of intensity for at least 6 min Spirometry after 5, 10, 15 and 30 min (additional measurements depending on patient’s clinical history)
**Diagnosis**
Mild EIB: FEV_1_ decrease ≥10% but <25% Moderate EIB: FEV_1_ decrease >25% but <50% Severe EIB: FEV_1_ decrease ≥50% (for steroid-naïve patients decrease in FEV_1_ ≥ 50%); for steroid-treated patients decrease in FEV_1_ ≥30%)

FEV_1_, forced expiratory volume in the first second; EIB, exercise-induced bronchoconstriction.

**Table 3 jcm-13-04558-t003:** Contraindications to test and medications and foods to be avoided before the challenge.

Contraindications
Dyspnoea at rest, FEV_1_ < 75% SpO_2_ < 85% in air Fever Pericarditis, myocarditis, uncontrolled hypertension, uncontrolled heart failure Acute kidney disease, acute hepatitis, uncontrolled diabetes Recent pneumothorax/pneumomediastinum, recent thoracic surgery
**Medications and foods to avoid before the test**
SABA (8 h) LABA (24 h) Anticholinergic drugs (12 h) Antileukotrienes (24 h) Theophylline retard (48 h) Cetirizine (3 days) Nedocromil sodium (48 h) Cromoglycate sodium (8 h) Coffee, tea, cola, chocolate (12 h)

FEV_1_, forced expiratory volume in the first second; SABA, short-acting beta_2_ agonists; LABA, long-acting beta_2_ agonists.

**Table 4 jcm-13-04558-t004:** Differential diagnosis of EIB.

Causes	Alternative Diagnosis
*Respiratory*	- Asthma without EIB - EILO, dysfunctional breathing disorders - Vascular malformations (vascular rings, pulmonary arteriovenous malformations) - Tracheobronchomalacia - Infectious diseases - Foreign body inhalation - Tumors - Interstitial diseases
*Cardiac*	- Arrhythmias (EIVT, SVT) - Pericarditis/Myocarditis - Shunting, vascular malformations - Cardiomyopathies - Pulmonary hypertension - Valvular abnormalities
*Metabolic/* *Neuromuscular*	- Mitochondrial disorders, mitochondrial enzyme deficiencies - Storage diseases (glycogenosis, sphingolipidoses) - Motor neuron diseases - Myopathies - Myasthenia gravis spectrum disorder
*Psychological*	- Anxiety, hyperventilation syndrome
*Others*	- Physiologic limitation - Physical deconditioning - Obesity - Exercise-associated gastroesophageal reflux - Anemia - Severe pectus excavatum

EILO, exercise-induced laryngeal obstruction; EIVT, effort-induced ventricular; SVT, supraventricular tachycardia.

**Table 5 jcm-13-04558-t005:** Treatment options for EIB.

Treatment Options for EIB	
Pharmacological	Nonpharmacological
*First choice* Short-acting beta_2_ agonists (SABA) about 10–15 min pre-exercise.	*Behavior* Warm-up Perform warm up exercise 10–15 min before the activity (for example interval high-intensity or a combination of high-intensity and low-intensity activity). Cooling down slowly for 5–10 min after exercise to prevent asthma symptoms after exercise.
*Second choice* In case of EIB despite using SABA or frequent use of SABA, consider: -inhaled corticosteroid (ICS), daily -leukotriene receptor antagonist (LTRA), daily	*Triggers avoidance* -Avoid exposure to cold air or wear a filter (such as a scarf or mask) and breathe through the nose during exercise. -Avoid outdoor exercise when pollution levels are very high. It is recommended to check the air quality indices online. -In case of allergy to pollen or mold it is useful to control environmental levels (through dedicated bulletins) avoiding outdoor exercise when they are very high. -Indoor training is recommended if outdoor trigger factors are present; alternatively, it is suggested to change the duration and intensity of outdoor training.
*Other alternatives* -Mast cell stabilizing agents pre-exercise -Inhaled anticholinergic agents pre-exercise	
Special population In asthmatic patients already treated with ICS/long acting beta_2_ antagonists (LABA), its use as needed before exercise is recommended (for example, Budesonide/Formoterol).	Lifestyle -Avoid smoking (both active and passive). -Follow a healthy diet, preferring food rich in vitamin D and omega 3 fatty acids and low in salt. -Practice regular aerobic exercise (at least 60 min of moderate to vigorous aerobic activity per day).

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
