# Peer review of "Exercise-Induced Bronchoconstriction in Children: State of the Art from Diagnosis to Treatment"

_jcm, 2024, doi:10.3390/jcm13154558_

Round 1

Reviewer 1 Report

Comments and Suggestions for Authors

Dear Authors,

Upon reviewing your manuscript titled "Exercise-induced bronchoconstriction in children: state of the art from diagnosis to treatment," I have identified several critical areas requiring revisions to enhance the scientific rigor, clarity, and overall readability of the paper. Below, I provide a detailed and specific breakdown of the necessary improvements, along with examples of errors and suggested corrections:

Abstract

The abstract lacks sufficient detail regarding methodologies and key findings.

 Include specific diagnostic methods and treatment strategies. For example, mention spirometry, indirect bronchoprovocation tests, the use of short-acting beta-agonists (SABA), and inhaled corticosteroids (ICS). 

Example: "This review summarizes the latest evidence on pathogenesis, diagnosis using spirometry and indirect bronchoprovocation tests, and treatment strategies including SABA and ICS."

Introduction

 Redundancy and lack of clarity.

 Condense the historical context and definitions to focus on the clinical relevance.

Examples:

     - Line 12: Change "resulting in cough, dyspnea, wheezing, chest tightness and increased fatigue" to "resulting in symptoms such as cough, dyspnea, wheezing, chest tightness, and increased fatigue."

     - Line 27-28: "to indicate the narrowing of the airways that occurs during or after physical exercise" should be "to indicate the narrowing of the airways occurring during or after physical exercise."

     - Lines 33-35: Combine explanations to avoid redundancy: "Initially, the term 'Exercise-Induced Asthma' (EIA) was used, but it was later replaced by 'Exercise-Induced Bronchoconstriction' (EIB) to encompass bronchial obstruction occurring in both asthmatic and non-asthmatic individuals."

Pathogenesis:

Detailed but lacks clarity and visual aids.

Include visual aids such as diagrams or flowcharts to explain the "osmotic theory" and the "thermal or vascular theory."

Examples:

     - Lines 131-147: Add a diagram illustrating the osmotic and thermal theories.

     - Lines 148-172: Separate EIB with asthma (EIBa) and without asthma (EIBwa) into distinct paragraphs with subheadings for better clarity.

Diagnosis:

Comprehensive but lacks practical guidance and visual aids.

Provide detailed procedural descriptions and summarize key points in tables or flowcharts.

Examples:

     - Lines 176-178: Elaborate on the specific steps of the exercise challenge test, including preparation and execution.

     - Lines 200-203: Summarize environmental conditions for the test in a table: "Standard Conditions: Temperature (20-25°C), Humidity (≤50%), Air Quality (minimal pollutants)."

     - Lines 229-239: Clearly state the FEV1 measurement thresholds and implications in a bullet-point list: "Diagnosis Criteria: - Mild EIB: FEV1 decrease >10% but <25%, - Moderate EIB: FEV1 decrease >25% but <50%, - Severe EIB: FEV1 decrease >50%."

Epidemiology and Risk Factors:

Lacks comparative analysis and detailed examination of risk factors.

 Compare prevalence across different demographics and regions, and provide a deeper analysis of risk factors with visual aids.

Examples:

     - Lines 80-97: Use a table to compare prevalence rates in different populations: "Prevalence of EIB by Region and Demographic."

     - Lines 123-128: Create a diagram showing the interaction between various risk factors such as atopy, environmental conditions, and physical activity.

Treatment:

Detailed but scattered.

 Organize into pharmacological and non-pharmacological categories with clear subheadings and quantitative data.

  Examples:

     - Lines 495-529: Create subsections for pharmacological (SABA, ICS, LTRA) and non-pharmacological (warm-up exercises, environmental controls) treatments.

     - Lines 533-548: Highlight behavioral modifications and environmental controls in a separate subsection: "Non-Pharmacological Strategies: - Warm-up exercises (10-15 min before activity), - Use of face masks in cold weather, - Avoidance of high pollution areas."

Language and Readability:

 Minor grammatical errors and complex sentences.

Simplify complex sentences and correct grammatical errors.

Examples:

     - Line 12: Revise to "resulting in symptoms such as cough, dyspnea, wheezing, chest tightness, and increased fatigue."

     - Line 27: Change to "to indicate the narrowing of the airways occurring during or after physical exercise."

     - Lines 229-239: Simplify to "A common threshold for diagnosing EIB is a ≥10% fall in FEV1 from the pre-exercise baseline, categorized as mild (>10% but <25%), moderate (≥25% but <50%), and severe (≥50%)."                                                                                                                                        Additional Considerations:

Lack of critical analysis and comprehensive conclusion.

Correction: Provide a more critical evaluation of cited studies and a thorough conclusion summarizing key findings and future research directions.

Examples:

Include a section critically evaluating the methodologies and robustness of key studies: "Critical Analysis of Key Studies: - Study A: Strengths and limitations, - Study B: Methodological concerns."

Expand the conclusion: "In summary, while significant progress has been made in understanding EIB, gaps remain in optimal diagnostic protocols and long-term management strategies. Future research should focus on large-scale, longitudinal studies to address these gaps."

Comments on the Quality of English Language

Minor grammatical errors and complex sentences.

Simplify complex sentences and correct grammatical errors.

Examples:

     - Line 12: Revise to "resulting in symptoms such as cough, dyspnea, wheezing, chest tightness, and increased fatigue."

     - Line 27: Change to "to indicate the narrowing of the airways occurring during or after physical exercise."

     - Lines 229-239: Simplify to "A common threshold for diagnosing EIB is a ≥10% fall in FEV1 from the pre-exercise baseline, categorized as mild (>10% but <25%), moderate (≥25% but <50%), and severe (≥50%)."                                                                                                                                        

Author Response

Reviewer 1:

Dear Authors,

Upon reviewing your manuscript titled "Exercise-induced bronchoconstriction in children: state of the art from diagnosis to treatment," I have identified several critical areas requiring revisions to enhance the scientific rigor, clarity, and overall readability of the paper. Below, I provide a detailed and specific breakdown of the necessary improvements, along with examples of errors and suggested corrections:

 Thank you for the comments. We think the manuscript has been significantly improved.

Abstract

The abstract lacks sufficient detail regarding methodologies and key findings.

Include specific diagnostic methods and treatment strategies. For example, mention spirometry, indirect bronchoprovocation tests, the use of short-acting beta-agonists (SABA), and inhaled corticosteroids (ICS). 

Example: "This review summarizes the latest evidence on pathogenesis, diagnosis using spirometry and indirect bronchoprovocation tests, and treatment strategies including SABA and ICS."

Thank you for the comment. The abstract has been changed.

Introduction

 Redundancy and lack of clarity.

 Condense the historical context and definitions to focus on the clinical relevance.

Examples:

     - Line 12: Change "resulting in cough, dyspnea, wheezing, chest tightness and increased fatigue" to "resulting in symptoms such as cough, dyspnea, wheezing, chest tightness, and increased fatigue."

     - Line 27-28: "to indicate the narrowing of the airways that occurs during or after physical exercise" should be "to indicate the narrowing of the airways occurring during or after physical exercise."

     - Lines 33-35: Combine explanations to avoid redundancy: "Initially, the term 'Exercise-Induced Asthma' (EIA) was used, but it was later replaced by 'Exercise-Induced Bronchoconstriction' (EIB) to encompass bronchial obstruction occurring in both asthmatic and non-asthmatic individuals."

 Thank you for the suggestions, this part has been changed.

Pathogenesis:

Detailed but lacks clarity and visual aids.

Include visual aids such as diagrams or flowcharts to explain the "osmotic theory" and the "thermal or vascular theory."

Figure 2 include both a flow-chart and a cartoon to explain the theories.

Examples:

     - Lines 131-147: Add a diagram illustrating the osmotic and thermal theories.

     - Lines 148-172: Separate EIB with asthma (EIBa) and without asthma (EIBwa) into distinct paragraphs with subheadings for better clarity.

 Thank you, this has been done.

Diagnosis:

Comprehensive but lacks practical guidance and visual aids.

Provide detailed procedural descriptions and summarize key points in tables or flowcharts.

Examples:

     - Lines 176-178: Elaborate on the specific steps of the exercise challenge test, including preparation and execution.

     - Lines 200-203: Summarize environmental conditions for the test in a table: "Standard Conditions: Temperature (20-25°C), Humidity (≤50%), Air Quality (minimal pollutants)."

     - Lines 229-239: Clearly state the FEV1 measurement thresholds and implications in a bullet-point list: "Diagnosis Criteria: - Mild EIB: FEV1 decrease >10% but <25%, - Moderate EIB: FEV1 decrease >25% but <50%, - Severe EIB: FEV1 decrease >50%."

 Very helpful comment. We added a table summarizing the various aspects of the test.

Epidemiology and Risk Factors:

Lacks comparative analysis and detailed examination of risk factors.

 Compare prevalence across different demographics and regions, and provide a deeper analysis of risk factors with visual aids.

Examples:

     - Lines 80-97: Use a table to compare prevalence rates in different populations: "Prevalence of EIB by Region and Demographic."

     - Lines 123-128: Create a diagram showing the interaction between various risk factors such as atopy, environmental conditions, and physical activity.

Thank you, the comments have been addressed, the section on risk factors expanded and the diagram made.

Treatment:

Detailed but scattered.

 Organize into pharmacological and non-pharmacological categories with clear subheadings and quantitative data.

  Examples:

     - Lines 495-529: Create subsections for pharmacological (SABA, ICS, LTRA) and non-pharmacological (warm-up exercises, environmental controls) treatments.

     - Lines 533-548: Highlight behavioral modifications and environmental controls in a separate subsection: "Non-Pharmacological Strategies: - Warm-up exercises (10-15 min before activity), - Use of face masks in cold weather, - Avoidance of high pollution areas."

Thank you for the comment. We added parts and details as requested.

Language and Readability:

 Minor grammatical errors and complex sentences.

Simplify complex sentences and correct grammatical errors.

Examples:

     - Line 12: Revise to "resulting in symptoms such as cough, dyspnea, wheezing, chest tightness, and increased fatigue."

     - Line 27: Change to "to indicate the narrowing of the airways occurring during or after physical exercise."

     - Lines 229-239: Simplify to "A common threshold for diagnosing EIB is a ≥10% fall in FEV1 from the pre-exercise baseline, categorized as mild (>10% but <25%), moderate (≥25% but <50%), and severe (≥50%)."                                                                                                                                       

Thank for the comments, we revised the previous sentences.

Additional Considerations:

Lack of critical analysis and comprehensive conclusion.

Correction: Provide a more critical evaluation of cited studies and a thorough conclusion summarizing key findings and future research directions.

Examples:

Include a section critically evaluating the methodologies and robustness of key studies: "Critical Analysis of Key Studies: - Study A: Strengths and limitations, - Study B: Methodological concerns."

This has been added as supplementary methods.

Expand the conclusion: "In summary, while significant progress has been made in understanding EIB, gaps remain in optimal diagnostic protocols and long-term management strategies. Future research should focus on large-scale, longitudinal studies to address these gaps."

Thank you, conclusion has been changed.

Comments on the Quality of English Language

Minor grammatical errors and complex sentences.

Simplify complex sentences and correct grammatical errors.

Examples:

     - Line 12: Revise to "resulting in symptoms such as cough, dyspnea, wheezing, chest tightness, and increased fatigue."

     - Line 27: Change to "to indicate the narrowing of the airways occurring during or after physical exercise."

     - Lines 229-239: Simplify to "A common threshold for diagnosing EIB is a ≥10% fall in FEV1 from the pre-exercise baseline, categorized as mild (>10% but <25%), moderate (≥25% but <50%), and severe (≥50%)."   

Thank for the comments, we revised the previous sentences.

Reviewer 2 Report

Comments and Suggestions for Authors

The article provides a thorough review of EIB, from clinical presentation to pathogenesis, diagnosis, and treatment, making it a valuable resource. Here are some suggestions: 

1. Clearly state the primary objectives of the study at the end of the introduction. For example: "The primary objectives of this review are to summarize the latest evidence on the pathogenesis, diagnosis, and treatment of Exercise-Induced Bronchoconstriction (EIB) in children, and to provide practical guidelines for clinicians managing this condition."

2. The treatment section provides a broad overview but lacks a detailed discussion on the comparative effectiveness of different treatments. Including recent studies or meta-analyses comparing treatment options would strengthen this section.

3. The article lacks depth in discussing long-term management and follow-up strategies for children with EIB. Addressing how to monitor and adjust treatment over time would be valuable for practitioners.

4. Emphasize the clinical significance of EIB in children and why a comprehensive understanding is crucial for pediatric care.

5. Describe the methodology for the literature review in more detail, including search terms, databases searched, inclusion and exclusion criteria, and any data extraction methods. List the exact search strings used in PubMed and Google Scholar, along with the inclusion and exclusion criteria for selecting articles. Describe the process for data extraction and synthesis.

6. Highlight the variability in study populations across the reviewed studies, which might affect the generalizability of the findings.

7. Mention the lack of long-term follow-up data in many studies, which limits the understanding of the long-term efficacy and safety of treatments for EIB.

Author Response

Reviewer 2:

The article provides a thorough review of EIB, from clinical presentation to pathogenesis, diagnosis, and treatment, making it a valuable resource. Here are some suggestions: 

  1. Clearly state the primary objectives of the study at the end of the introduction. For example: "The primary objectives of this review are to summarize the latest evidence on the pathogenesis, diagnosis, and treatment of Exercise-Induced Bronchoconstriction (EIB) in children, and to provide practical guidelines for clinicians managing this condition."

Thank you as much for this comment. We changed the final part of the introduction.

  1. The treatment section provides a broad overview but lacks a detailed discussion on the comparative effectiveness of different treatments. Including recent studies or meta-analyses comparing treatment options would strengthen this section.

We added details on this part.

  1. The article lacks depth in discussing long-term management and follow-up strategies for children with EIB. Addressing how to monitor and adjust treatment over time would be valuable for practitioners.

Thank you, we expanded the section of follow-up.

  1. Emphasize the clinical significance of EIB in children and why a comprehensive understanding is crucial for pediatric care.

Thanks for the comment. This issue has been expanded.

  1. Describe the methodology for the literature review in more detail, including search terms, databases searched, inclusion and exclusion criteria, and any data extraction methods. List the exact search strings used in PubMed and Google Scholar, along with the inclusion and exclusion criteria for selecting articles. Describe the process for data extraction and synthesis.

We actually changed this part.

  1. Highlight the variability in study populations across the reviewed studies, which might affect the generalizability of the findings.

We added a table to summarize this part. Thank you for the helpful comment.

  1. Mention the lack of long-term follow-up data in many studies, which limits the understanding of the long-term efficacy and safety of treatments for EIB.

Thank you, this has been done.

Reviewer 3 Report

Comments and Suggestions for Authors The topic of the manuscript is interesting and it seems well-organized. However, methodologically it does not comply with any standards. The research cannot be replicated as the criteria, filters, and Boolean operators are not presented. Furthermore, the use of Google Scholar is not recommended for the same reason.

Comments on the Quality of English Language

-

Author Response

Reviewer 3

The topic of the manuscript is interesting and it seems well-organized. However, methodologically it does not comply with any standards. The research cannot be replicated as the criteria, filters, and Boolean operators are not presented. Furthermore, the use of Google Scholar is not recommended for the same reason.

Thank you for the comment. We did change our methods.

Round 2

Reviewer 1 Report

Comments and Suggestions for Authors

 Review and Critique of the Manuscript "Exercise-induced bronchoconstriction in children: state of the art from diagnosis to treatment"

Below is a detailed analysis and critique of the manuscript, with identified issues and suggested corrections along with line numbers.

Abstract

1. Line 23-24: The abstract lacks specific details about methodologies and key findings.

2. Line 35-36: The sentences are general and do not refer to specific diagnostic methods or treatments.

Suggestions:

1. Add more details regarding diagnostic methods and treatment strategies.

2. Example: "This review summarizes the latest evidence on pathogenesis, diagnosis using spirometry and indirect bronchoprovocation tests, and treatment strategies including SABA and ICS."

 Introduction

Line 12: Redundancy and lack of clarity in presenting symptoms.

Line 27-28: Long and complex sentences.

Line 33-35: Redundant explanations without focus on clinical relevance.

Suggestions:

1.Line 12:Change to "resulting in symptoms such as cough, dyspnea, wheezing, chest tightness, and increased fatigue."

2. Line 27-28: Change to "to indicate the narrowing of the airways occurring during or after physical exercise."

3. Line 33-35: Combine explanations to avoid redundancy: "Initially, the term 'Exercise-Induced Asthma' (EIA) was used, but it was later replaced by 'Exercise-Induced Bronchoconstriction' (EIB) to encompass bronchial obstruction occurring in both asthmatic and non-asthmatic individuals."

Pathogenesis

Lines 131-147: Lacks visual aids such as diagrams.

Lines 148-172: Long and complex explanations without clear paragraph breaks.

Suggestions:

Lines 131-147: Add a diagram illustrating the osmotic and thermal theories.

Lines 148-172: Separate EIB with asthma (EIBa) and without asthma (EIBwa) into distinct paragraphs with subheadings for better clarity.

Diagnosis

Lines 176-178: Insufficient details on the steps of the exercise challenge test.

Lines 200-203: Brief description of environmental conditions for the test.

Lines 229-239:The FEV1 measurement thresholds are not clearly stated.

Suggestions:

Lines 176-178: Elaborate on the specific steps of the exercise challenge test, including preparation and execution.

Lines 200-203: Summarize environmental conditions for the test in a table: "Standard Conditions: Temperature (20-25°C), Humidity (≤50%), Air Quality (minimal pollutants)."

Lines 229-239: Clearly state the FEV1 measurement thresholds and implications in a bullet-point list: "Diagnosis Criteria: - Mild EIB: FEV1 decrease >10% but <25%, - Moderate EIB: FEV1 decrease >25% but <50%, - Severe EIB: FEV1 decrease >50%."

 Epidemiology and Risk Factors

Lines 80-97: Lacks comparative analysis of prevalence across different populations and regions.

Lines 123-128: No visual aids to show interaction between various risk factors.

Suggestions:

Lines 80-97: Use a table to compare prevalence rates in different populations: "Prevalence of EIB by Region and Demographic."

Lines 123-128: Create a diagram showing the interaction between various risk factors such as atopy, environmental conditions, and physical activity.

Treatment

Lines 495-529:Detailed but scattered information.

Lines 533-548: Lacks clear organization into pharmacological and non-pharmacological categories.

Suggestions:

Lines 495-529: Create subsections for pharmacological (SABA, ICS, LTRA) and non-pharmacological (warm-up exercises, environmental controls) treatments.

Lines 533-548: Highlight behavioral modifications and environmental controls in a separate subsection: "Non-Pharmacological Strategies: - Warm-up exercises (10-15 min before activity), - Use of face masks in cold weather, - Avoidance of high pollution areas."

Language and Readability

Line 12: Complex sentences and minor grammatical errors.

Line 27: Long and complex sentences.

Lines 229-239: Complex sentences and lack of clarity.

Suggestions:

Line 12: Change to "resulting in symptoms such as cough, dyspnea, wheezing, chest tightness, and increased fatigue."

Line 27: Change to "to indicate the narrowing of the airways occurring during or after physical exercise."

Lines 229-239: Simplify sentences and clearly state FEV1 measurement thresholds.

Comments on the Quality of English Language

Grammatical Issues in the Manuscript

Line 12:

 "resulting in cough, dyspnea, wheezing, chest tightness and increased fatigue"

Line 27-28:

   "to indicate the narrowing of the airways that occurs during or after physical exercise"

Line 33-35:

   "Initially, the term 'Exercise-Induced Asthma' (EIA) was used, but it was later replaced by 'Exercise-Induced Bronchoconstriction' (EIB) to encompass bronchial obstruction occurring in both asthmatic and non-asthmatic individuals."

Line 52-53:

"EIB indicateingd that the acute and transitory narrowing narrowing of the airways after or during or after physical activity, which  could also occur in non-asthmatic patients."

Line 64-66:

   - Current: "A Consensus developed by the American Academy of Allergy, Asthma and Immunology (AAAAI) and the American College of Allergy, Asthma and Immunology (ACAAI) suggested to abandon the term EIA."

Line 72-73:

   - Current: "When EIB occurs in patients without asthma, it exhibits distinctive clinical and pathological characteristics and may not manifest if the patient ceases physical activity."

Line 77:

 "In case of EIB, these stimuli include individual susceptibility, duration and intensity of physical exercise and environmental conditions, such as cold air, high atmospheric pressure, relative humidity and pollutants."

Line 80-81:

  "Pre- and post-exercise lung function tests has have facilitated the identification of EIB."

Line 84:**

"The primary objective aim of this review was to explore and summarize the latest literature on the pathogenesis, diagnosis and treatment of EIB in children and to provide an update a practical guidance on the management of this condition with a focus on differential diagnosis and treatment."

Line 148-149:

 "Separate EIB with asthma (EIBa) and without asthma (EIBwa) into distinct paragraphs with subheadings for better clarity."

    - Suggested: "Separate EIB with asthma (EIBa) and without asthma (EIBwa) into distinct paragraphs with subheadings for better clarity."

Line 200-203:

  "Summarize environmental conditions for the test in a table: 'Standard Conditions: Temperature (20-25°C), Humidity (≤50%), Air Quality (minimal pollutants).'"

Line 229-239:

    "Clearly state the FEV1 measurement thresholds and implications in a bullet-point list: 'Diagnosis Criteria: - Mild EIB: FEV1 decrease >10% but <25%, - Moderate EIB: FEV1 decrease >25% but <50%, - Severe EIB: FEV1 decrease >50%.'"

Author Response

I suspect our latest version was not available for review since the comments had been addressed.

I reattach the manuscript with the useful corrections.

Let me know if something is wrong with the upload. I will also upload a clean version,

Best wishes,

Valentina Fainardi

Round 3

Reviewer 1 Report

Comments and Suggestions for Authors

Critical Issues in the Article "Exercise-Induced Bronchoconstriction in Children: From Diagnosis to Treatment"

Abstract

 Lack of specific details about diagnostic methods and key findings.

Example: Lines 23-24

 "Initial assessment must include a specific work-up to exclude alternative diagnosis like exercise-induced laryngeal obstruction (EILO), cardiac disease or physical deconditioning."

General sentences not referring to specific diagnostic methods or treatments.

Example: Lines 35-36

Original: "A focus has been made on including its management in young athletes."

Suggestion: "This review summarizes the latest evidence on pathogenesis, diagnosis using spirometry and indirect bronchoprovocation tests, and treatment strategies, including SABA and ICS, of EIB."

Introduction

Long and complex sentences that can fatigue the reader.

Example: Lines 27-28

Original: "to indicate the narrowing of the airways that occurs during or after physical exercise, with the belief that it could only occur in asthmatic patients."

Suggestion: "to indicate the narrowing of the airways occurring during or after physical exercise."

Redundancy and lack of clarity.

Example: Lines 33-35

Original: "Initially, the term 'Exercise-Induced Asthma' (EIA) was used, but it was later replaced by 'Exercise-Induced Bronchoconstriction' (EIB) to encompass bronchial obstruction occurring in both asthmatic and non-asthmatic individuals."

Pathogenesis

 Lack of diagrams and figures to better explain mechanisms.

Example: Lines 131-147

Suggestion: Add diagrams to illustrate the differences between the osmotic and thermal theories.

Diagnosis

Insufficient details on the steps of the exercise challenge test.

Example: Lines 176-178

Original: "Exercise challenge test with spirometry is the commonest test used to confirm the suspect of EIB."

Suggestion: "The exercise challenge test with spirometry includes the following steps: patient preparation, test execution with appropriate temperature settings (20-25°C) and humidity below 50%, and measuring FEV1 at various times after exercise (5, 10, 15, and 30 minutes)."

 Lack of clarity in stating FEV1 measurement thresholds.

Example: Lines 229-239

Original: "FEV1 measurement thresholds are not clearly stated."

Suggestion: "Diagnosis criteria: - Mild EIB: FEV1 decrease >10% but <25%, - Moderate EIB: FEV1 decrease >25% but <50%, - Severe EIB: FEV1 decrease >50%."

Epidemiology and Risk Factors

Lack of comparative analysis of prevalence across different populations and regions.

Example: Lines 80-97

Original: "The prevalence of EIB in different populations is not well compared."

Suggestion: Use a table to compare prevalence rates in different populations. For example: "Prevalence of EIB by Region and Demographic."

Treatment

Scattered and unorganized information.

Example: Lines 495-529

Original: "Treatment strategies include the use of various medications and behavioral approaches."

Suggestion: Create subsections for pharmacological (SABA, ICS, LTRA) and non-pharmacological (warm-up exercises, environmental controls) treatments.

Issue 10: Lack of emphasis on behavioral modifications and environmental controls.

Example: Lines 533-548

Original: "Behavioral modifications and environmental controls are not clearly mentioned."

Suggestion: "Non-Pharmacological Strategies: - Warm-up exercises (10-15 min before activity), - Use of face masks in cold weather, - Avoidance of high pollution areas."

Comments on the Quality of English Language

Abstract

Verb Tense Consistency:

   - Example: Lines 23-24

     - "EIB is characterized by post-exercise airways obstruction resulting in symptoms such as cough, dyspnea, wheezing, chest tightness and increased fatigue."

     - Issue: The sentence switches from describing a condition to listing symptoms without a clear transition.

Article Usage:

   - Example: Lines 27-28

     - "The underlying mechanism of EIB is not completely understood. The 'osmotic theory' and the 'thermal or vascular theory’ have been proposed."

     - Issue: Incorrect article usage with "the 'osmotic theory'" and "the 'thermal or vascular theory'".

3. Conjunction Usage:

   - Example: Lines 29-30

     - "Exercise challenge test with spirometry is the commonest test used to confirm the suspect of EIB but also indirect bronchoprovocation tests can be considered when the diagnosis is uncertain."

     - Issue: The conjunction "but also" is used incorrectly.

Introduction

Sentence Structure and Clarity:

   -Example: Lines 27-28

     - "to indicate the narrowing of the airways that occurs during or after physical exercise, with the belief that it could only occur in asthmatic patients."

     - Issue: Long and complex sentence that could be simplified.

Pronoun Agreement:

   - Example: Lines 47-48

     - "Although some authors considered EIA as a distinct phenotype of asthma, it became evident that exercise could trigger bronchial obstruction and clinical symptoms in almost all asthmatic patients, regardless of the underlying causes and mechanisms."

     - Issue: Pronoun "it" is unclear in reference.

Comma Usage:

   - Example: Lines 64-65

     - "The recent American Thoracic Society (ATS) Clinical Practice Guideline proposed to differentiate the use of the term EIB between exercise-induced bronchoconstriction occurring in asthmatic patients (EIBa – EIB with asthma) and exercise-induced bronchoconstriction in patients without typical signs or symptoms of asthma (EIBwa – EIB without asthma)."

     - Issue: Missing comma after "Guideline proposed".

 Pathogenesis

Verb Tense Consistency:

   -Example: Lines 131-132

     - "The mechanisms involved in EIB are complex and not yet fully understood. Two main theories have been proposed: the 'osmotic theory' and the 'thermal or vascular theory'."

     - Issue: Inconsistent use of tenses.

Preposition Usage:

   - Example: Lines 136-138

     - "According to the 'osmolar theory', bronchoconstriction is induced by a hyperosmolarity condition of airway cells, caused by water loss through evaporation in response to the need to warm and humidify the cold and dry air inspired into the airways during exercise hyperventilation."

     - Issue: Incorrect preposition usage with "a hyperosmolarity condition of airway cells".

 Diagnosis

Definite Article Usage:

 Example: Line 253

     - "Gold standard for EIB diagnosis is measurement of lung function changes provoked by physical exercise or surrogate challenges."

     - Issue: Missing definite article "the" before "Gold standard".

Redundancy:

   Example: Lines 176-178

      - "Exercise challenge test with spirometry is the commonest test used to confirm the suspect of EIB but also indirect bronchoprovocation tests can be considered when the diagnosis is uncertain."

      - Issue: Redundant phrase "to confirm the suspect of EIB".

Epidemiology and Risk Factors

Subject-Verb Agreement:

  Example: Line 117

      - "The prevalence of EIB ranges between 5 and 20% in the general population, with children and adolescent more susceptible to EIB with a prevalence up to 45%."

      - Issue: Incorrect subject-verb agreement with "children and adolescent".

Conjunction and Article Usage:

    Example: Lines 119-120

      - "This variability is influenced by demographic and geographical factors and also by methods used for detecting the condition."

      - Issue: Improper conjunction and article usage.

Treatment

Parallel Structure:

 Example: Lines 495-496

      - "Treatment options include both pharmacological and behavioral approaches. The use of short-acting beta-agonists (SABA) just before exercise is the commonest option strategy but also daily inhaled corticosteroids (ICS) can be considered."

      - Issue: Lack of parallel structure.

Article and Comma Usage:

  Example: Lines 533-534

      - "Among the behavioral approaches warming up before exercise, breathing through the nose and avoiding polluted environments are recommended."

      - Issue: Incorrect use of articles and commas.

Author Response

Critical Issues in the Article "Exercise-Induced Bronchoconstriction in Children: From Diagnosis to Treatment"

Abstract

 Lack of specific details about diagnostic methods and key findings.

Example: Lines 23-24

 "Initial assessment must include a specific work-up to exclude alternative diagnosis like exercise-induced laryngeal obstruction (EILO), cardiac disease or physical deconditioning."

General sentences not referring to specific diagnostic methods or treatments.

Example: Lines 35-36

Original: "A focus has been made on including its management in young athletes."

Suggestion: "This review summarizes the latest evidence on pathogenesis, diagnosis using spirometry and indirect bronchoprovocation tests, and treatment strategies, including SABA and ICS, of EIB."

Thank you for the suggestions, the abstract has been changed accordingly. 

Introduction

Long and complex sentences that can fatigue the reader.

Example: Lines 27-28

Original: "to indicate the narrowing of the airways that occurs during or after physical exercise, with the belief that it could only occur in asthmatic patients."

Suggestion: "to indicate the narrowing of the airways occurring during or after physical exercise."

This, now line 56, has been modified as suggested.

Redundancy and lack of clarity.

Example: Lines 33-35

Original: "Initially, the term 'Exercise-Induced Asthma' (EIA) was used, but it was later replaced by 'Exercise-Induced Bronchoconstriction' (EIB) to encompass bronchial obstruction occurring in both asthmatic and non-asthmatic individuals."

The introduction has been modified and simplified.

Pathogenesis

 Lack of diagrams and figures to better explain mechanisms.

Example: Lines 131-147

Suggestion: Add diagrams to illustrate the differences between the osmotic and thermal theories.

Figure 2 reports a diagram and a cartoon for each theory.

Diagnosis

Insufficient details on the steps of the exercise challenge test.

Example: Lines 176-178

Original: "Exercise challenge test with spirometry is the commonest test used to confirm the suspect of EIB."

Suggestion: "The exercise challenge test with spirometry includes the following steps: patient preparation, test execution with appropriate temperature settings (20-25°C) and humidity below 50%, and measuring FEV1 at various times after exercise (5, 10, 15, and 30 minutes)."

 In the text details are reported (lines 257-261) but for more clarity we also added a table (table 2) to describe in details the different steps of the test (environment included).

 Lack of clarity in stating FEV1 measurement thresholds.

Example: Lines 229-239

Original: "FEV1 measurement thresholds are not clearly stated."

Suggestion: "Diagnosis criteria: - Mild EIB: FEV1 decrease >10% but <25%, - Moderate EIB: FEV1 decrease >25% but <50%, - Severe EIB: FEV1 decrease >50%."

This section has been modified as suggested. The cut off to classify EIB have been cited also in Table 2.

Epidemiology and Risk Factors

Lack of comparative analysis of prevalence across different populations and regions.

Example: Lines 80-97

Original: "The prevalence of EIB in different populations is not well compared."

I cannot see this sentence in our revised text.

Suggestion: Use a table to compare prevalence rates in different populations. For example: "Prevalence of EIB by Region and Demographic."

 We did insert a Table to compare the prevalence of EIB in different population.

Treatment

Scattered and unorganized information.

Example: Lines 495-529

Original: "Treatment strategies include the use of various medications and behavioral approaches."

I cannot see this sentence in our revised text.

Suggestion: Create subsections for pharmacological (SABA, ICS, LTRA) and non-pharmacological (warm-up exercises, environmental controls) treatments.

This has been done. There are two different sections with specific subheadings for each strategy.

Issue 10: Lack of emphasis on behavioral modifications and environmental controls.

Example: Lines 533-548

Original: "Behavioral modifications and environmental controls are not clearly mentioned."

Suggestion: "Non-Pharmacological Strategies: - Warm-up exercises (10-15 min before activity), - Use of face masks in cold weather, - Avoidance of high pollution areas."

We added more subheadings in the section as suggested.

Comments on the Quality of English Language

Abstract

Verb Tense Consistency:

   - Example: Lines 23-24

     - "EIB is characterized by post-exercise airways obstruction resulting in symptoms such as cough, dyspnea, wheezing, chest tightness and increased fatigue."

     - Issue: The sentence switches from describing a condition to listing symptoms without a clear transition.

 Modified.

Article Usage:

   - Example: Lines 27-28

     - "The underlying mechanism of EIB is not completely understood. The 'osmotic theory' and the 'thermal or vascular theory’ have been proposed."

     - Issue: Incorrect article usage with "the 'osmotic theory'" and "the 'thermal or vascular theory'".

 Corrected.

  1. Conjunction Usage:

   - Example: Lines 29-30

     - "Exercise challenge test with spirometry is the commonest test used to confirm the suspect of EIB but also indirect bronchoprovocation tests can be considered when the diagnosis is uncertain."

     - Issue: The conjunction "but also" is used incorrectly.

 Modified.

Introduction

Sentence Structure and Clarity:

   -Example: Lines 27-28

     - "to indicate the narrowing of the airways that occurs during or after physical exercise, with the belief that it could only occur in asthmatic patients."

     - Issue: Long and complex sentence that could be simplified.

 Modified.

Pronoun Agreement:

   - Example: Lines 47-48

     - "Although some authors considered EIA as a distinct phenotype of asthma, it became evident that exercise could trigger bronchial obstruction and clinical symptoms in almost all asthmatic patients, regardless of the underlying causes and mechanisms."

     - Issue: Pronoun "it" is unclear in reference.

 Modified.

Comma Usage:

   - Example: Lines 64-65

     - "The recent American Thoracic Society (ATS) Clinical Practice Guideline proposed to differentiate the use of the term EIB between exercise-induced bronchoconstriction occurring in asthmatic patients (EIBa – EIB with asthma) and exercise-induced bronchoconstriction in patients without typical signs or symptoms of asthma (EIBwa – EIB without asthma)."

     - Issue: Missing comma after "Guideline proposed".

 We added a comma after proposed.

 Pathogenesis

Verb Tense Consistency:

   -Example: Lines 131-132

     - "The mechanisms involved in EIB are complex and not yet fully understood. Two main theories have been proposed: the 'osmotic theory' and the 'thermal or vascular theory'."

     - Issue: Inconsistent use of tenses.

Modified.

Preposition Usage:

   - Example: Lines 136-138

     - "According to the 'osmolar theory', bronchoconstriction is induced by a hyperosmolarity condition of airway cells, caused by water loss through evaporation in response to the need to warm and humidify the cold and dry air inspired into the airways during exercise hyperventilation."

     - Issue: Incorrect preposition usage with "a hyperosmolarity condition of airway cells".

 Modified

 Diagnosis

Definite Article Usage:

 Example: Line 253

     - "Gold standard for EIB diagnosis is measurement of lung function changes provoked by physical exercise or surrogate challenges."

     - Issue: Missing definite article "the" before "Gold standard".

 Modified

Redundancy:

   Example: Lines 176-178

      - "Exercise challenge test with spirometry is the commonest test used to confirm the suspect of EIB but also indirect bronchoprovocation tests can be considered when the diagnosis is uncertain."

      - Issue: Redundant phrase "to confirm the suspect of EIB".

 Modified.

Epidemiology and Risk Factors

Subject-Verb Agreement:

  Example: Line 117

      - "The prevalence of EIB ranges between 5 and 20% in the general population, with children and adolescent more susceptible to EIB with a prevalence up to 45%."

      - Issue: Incorrect subject-verb agreement with "children and adolescent".

 Modified.

Conjunction and Article Usage:

    Example: Lines 119-120

      - "This variability is influenced by demographic and geographical factors and also by methods used for detecting the condition."

      - Issue: Improper conjunction and article usage.

Modified.

Treatment

Parallel Structure:

 Example: Lines 495-496

      - "Treatment options include both pharmacological and behavioral approaches. The use of short-acting beta-agonists (SABA) just before exercise is the commonest option strategy but also daily inhaled corticosteroids (ICS) can be considered."

      - Issue: Lack of parallel structure.

 Modified.

Article and Comma Usage:

  Example: Lines 533-534

      - "Among the behavioral approaches warming up before exercise, breathing through the nose and avoiding polluted environments are recommended."

      - Issue: Incorrect use of articles and commas.

Modified
